# Association of Preoperative Prognostic Nutritional Index and Postoperative Acute Kidney Injury in Patients Who Underwent Hepatectomy for Hepatocellular Carcinoma

**DOI:** 10.3390/jpm11050428

**Published:** 2021-05-18

**Authors:** Ji Hoon Sim, In-Gu Jun, Young-Jin Moon, A Rom Jeon, Sung-Hoon Kim, Bomi Kim, Jun-Gol Song

**Affiliations:** Department of Anesthesiology and Pain Medicine, Asan Medical Center, University of Ulsan College of Medicine, Seoul 05505, Korea; atlassjh@hanmail.net (J.H.S.); igjun@amc.seoul.kr (I.-G.J.); yjmoon@amc.seoul.kr (Y.-J.M.); atomy0429@gmail.com (A.R.J.); shkimans@gmail.com (S.-H.K.); dudnd5@gmail.com (B.K.)

**Keywords:** acute kidney injury, hepatocellular carcinoma, prognostic nutritional index, mortality

## Abstract

Various biological indicators are reportedly associated with postoperative acute kidney injury (AKI) in the surgical treatment of hepatocellular carcinoma (HCC). However, only a few studies have evaluated the association between the preoperative prognostic nutritional index (PNI) and postoperative AKI. This study evaluated the association of the preoperative PNI and postoperative AKI in HCC patients. We retrospectively analyzed 817 patients who underwent open hepatectomy between December 2007 and December 2015. Multivariate regression analysis was performed to evaluate the association between the PNI and postoperative AKI. Additionally, we evaluated the association between the PNI and outcomes such as postoperative renal replacement therapy (RRT) and mortality. Cox regression analysis was performed to assess the risk factors for one-year and five-year mortality. In the multivariate analysis, high preoperative PNI was significantly associated with a lower incidence of postoperative AKI (odds ratio (OR): 0.92, 95% confidence interval (CI): 0.85 to 0.99, *p* = 0.021). Additionally, diabetes mellitus and the use of synthetic colloids were significantly associated with postoperative AKI. PNI was associated with postoperative RRT (OR: 0.76, 95% CI: 0.60 to 0.98, *p* = 0.032) even after adjusting for other potential confounding variables. In the Cox regression analysis, high PNI was significantly associated with low one-year mortality (Hazard ratio (HR): 0.87, 95% CI: 0.81 to 0.94, *p* < 0.001), and five-year mortality (HR: 0.93, 95% CI: 0.90–0.97, *p* < 0.001). High preoperative PNI was significantly associated with a lower incidence of postoperative AKI and low mortality. These results suggest that the preoperative PNI might be a predictor of postoperative AKI and surgical prognosis in HCC patients undergoing open hepatectomy.

## 1. Introduction

Hepatectomy is a major intra-abdominal general surgery with a high risk of postoperative AKI, which is unique due to the patient’s underlying disease, as well as surgical and anesthetic management considerations [1]. The reported incidence of AKI in patients who underwent hepatic resection for HCC is 15% to 20% [2,3,4]. AKI increases morbidity, mortality, and hospital costs [4,5,6]. AKI is also associated with a risk of developing chronic kidney disease (CKD) [7]. Therefore, there have been many studies to identify the risk factors for postoperative AKI in HCC patients [8,9,10,11,12,13]. However, until now, there have been few well-established risk factors for postoperative AKI in the context of hepatectomy.

The prognostic nutritional index (PNI), a biological marker, which can be determined through the serum albumin level and total lymphocyte count in peripheral blood, reflects a patient’s nutritional and inflammatory status [14]. The PNI was reportedly associated with survival or complications in various diseases, including cancer [15,16,17,18]. Recent studies have also indicated that the PNI is an effective predictor of prognosis in patients with HCC after hepatectomy [19,20].

However, only a few studies have reported the association between the preoperative PNI and postoperative AKI in HCC patients. Therefore, we aimed to evaluate the prognostic role of the preoperative PNI in developing postoperative AKI in HCC patients who had undergone hepatectomy. We also assessed surgical outcomes such as the length of hospital stay, postoperative intensive care unit (ICU) admission, and mortality rate.

## 2. Materials and Methods

### 2.1. Study Design & Patient Population

We retrospectively reviewed 817 patients diagnosed with HCC according to the 10th revision of the International Classification of Diseases (ICD-10) guidelines, who underwent liver resection between December 2007 and December 2015. The institutional review board of Asan Medical Center (protocol number: 2021-0243) approved this study and waived the need for written consent. Adult patients aged over 18 years who underwent open hepatectomy were included in the study. All surgeries were planned and performed by a single surgeon. The exclusion criteria were as follows: patients aged < 18 or ≥ 80 years, patients with severe cardiopulmonary or chronic kidney disease, patients who had already received renal replacement therapy (RRT), patients who required intervention in the urinary system during surgery, patients who underwent emergency surgery, and patients with incomplete data or missing PNI or serum creatinine values.

### 2.2. Anesthetic & Surgical Technique

For the induction of general anesthesia, we administered an intravenous bolus injection of thiopental sodium (4 to 5 mg/kg) or propofol (1 to 2 mg/kg). After loss of consciousness, rocuronium (0.6 to 1.2 mg/kg) was administered for muscle relaxation. Before tracheal intubation, 1 to 2 μg/kg of fentanyl was administered as an intravenous bolus injection, and maintenance of anesthesia was achieved with 2 to 4 vol% sevoflurane in 50% air/oxygen. After insertion of the arterial and central venous lines, invasive monitoring of arterial and central venous pressure was routinely performed. The patient was mechanically ventilated with a tidal volume of 6 to 8 mL/kg, and the end-expiratory carbon dioxide partial pressure was adjusted to maintain a value of 35 to 40 mmHg. During anesthesia, crystalline solutions (Ringer’s lactate solution or plasma solution) or colloidal solutions (5% albumin or synthetic colloids (Voluven^®^; Fresenius Kabi, Bad Homburg, Germany)) were administered, and the total volume of synthetic colloids did not exceed 20 mL/kg. When the plasma hemoglobin (Hb) level was less than 8 g/dL during surgery, packed red blood cell (RBC) transfusion was performed. The Hb level in patients with ischemic heart disease was maintained at >10 g/dL. Vasopressors such as ephedrine or phenylephrine were administered when the mean arterial blood pressure was less than 65 mmHg, and inotropic agents such as norepinephrine were administered when the vasopressor was ineffective, under the clinical judgment of an anesthesiologist. 

Open abdominal surgery was performed through a J-shaped incision. Liver parenchymal transection was performed using an ultrasonic surgical aspirator (CUSA^®^ Excel; Valleylab Inc., Boulder, CO, USA). After parenchymal dissection, the hepatic vein was cut and ligated with a stapler or clamp and then sutured. Liver resection was performed based on the definition by Couinaud classification [21]. Minor resection was defined when hepatic resection was limited to two or fewer segments, and the others were defined as a major resection [22]. Right anterior and posterior sectionectomy were classified as major resection as advanced techniques and a longer operation time are frequently required for those types of hepatic resections [23]. Hemostasis was achieved using electrocoagulation, clips, argon beam, or nonabsorbable sutures. Systematic placement of abdominal drainage was routinely performed during surgery.

After surgery, crystalloids and colloids were administered appropriately to maintain fluid balance and normal kidney function. Management with sodium restriction and judicious use of diuretic therapy were carried out when there was new onset postoperative ascites, and we did not use diuretics routinely. Blood tests were performed daily for three days after surgery to measure the patient’s complete blood count, biochemical and electrolyte levels, and inflammation levels. We routinely used postoperative intravenous patient-controlled analgesia (PCA) for pain control. In some cases, hepatoprotective agents such as branched-chain amino acids (BCAA) were administered to support protein synthesis and regeneration of the remnant liver.

### 2.3. Clinical Data Collection and Outcome Assessments

Demographic data and data on perioperative variables were collected from the electronic medical record system. Demographic data included age, sex, weight, body mass index, TNM staging, number of tumors, size of tumors, lymph node invasion, and distant metastasis. Data regarding the presence of comorbid diseases, such as diabetes mellitus (DM), hypertension, coronary artery disease, and cerebrovascular accident, liver cirrhosis, indocyanine green retention rate at 15 min (ICG R15), and the model for end-stage liver disease (MELD) and Child–Turcotte–Pugh scores, which indicate the severity of liver disease, were also collected.

Laboratory values included preoperative white blood cell count, Hb, platelet count, prothrombin time, albumin, serum creatinine (sCr), estimated glomerular filter rate (eGFR), total bilirubin, aspartate aminotransferase (AST), alanine aminotransferase (ALT), and sodium. Serum creatinine levels were checked daily, from postoperative day 1 to day 7, to confirm AKI. Preoperative PNI values were also collected. The PNI was calculated using the following formula: [10 × serum albumin (g/dL)] + [0.005 × total lymphocyte count (per mm^3^)]. The total blood counts of all patients were determined preoperatively, less than two days after admission, and prior to surgery. Intraoperative variables included operation time, total fluids, administered crystalloids and colloids, RBC transfusion, diuretics, and urine output. Data on postoperative AKI incidence and grade, postoperative RRT, post-hepatectomy liver failure (PHLF), hospital stay, ICU admission, prolonged ICU stay (≥two days), and one-year mortality (calculated from the date of surgery to one-year follow-up), and overall mortality (determined from the date of surgery to the last follow-up) records were also collected. 

### 2.4. Primary and Secondary Outcomes

The primary outcome was the association between postoperative AKI and the preoperative PNI. The secondary outcome was the association between the preoperative PNI and outcomes, such as postoperative RRT, PHLF, ICU admission, one-year, five-year, and overall mortality. Additionally, we evaluated the risk factors associated with postoperative AKI and one-year, and five-year mortality. The Kidney Disease Improving Global Outcomes classification defines AKI as an increase in sCr ≥ 1.5 times the baseline value, within seven days prior to surgery or increase in sCr by ≥0.3 mg/dL within 48 h [24]. PHLF was defined by the International Study Group of Liver Surgery (ISGLS) [25].

### 2.5. Statistical Analysis

Categorical data were analyzed using the chi-squared test or Fisher’s exact test, and continuous data were evaluated using the independent *t*-test or Mann–Whitney U test. The data are appropriately presented as mean ± standard deviation (SD), median of the quartile range, or numbers with proportions. We used the multivariate logistic regression analysis to determine the association between the preoperative PNI and AKI. In univariate analysis, all variables with *p*-values less than 0.1 were included in the multivariate analysis. Cox regression analysis was used to evaluate the adjusted risk ratio of one-year and five-year mortality. The proportional hazard assumption was tested using Schoenfeld’s residual test. We set the PNI cutoff value for mortality rate to 45, guided by receiver operating characteristic analysis (AUC: 0.731, sensitivity: 74.58, specificity: 61.21). The Kaplan–Meier method was used to analyze the cumulative one-year and five-year survival between the PNI < 45 and PNI ≥ 45 groups. The log-rank test was used to evaluate the change between curves. All *p*-values less than 0.05 were considered statistically significant. Data manipulation and statistical analyses were performed using the IBM SPSS Statistics for Windows, version 22.0 (IBM Corporation, Armonk, NY, USA).

## 3. Results

Of the 921 enrolled patients, 104 were excluded based on the study criteria. Finally, a total of 817 patients were enrolled in this study (Figure 1).

Table 1 shows the demographic data, perioperative variables, and surgical outcomes of the study population. The postoperative AKI incidence rate was 7.2% (59/817); most of the cases were grade 1, and grades 2 and 3 occurred in four and three patients, respectively. The percentage of patients who underwent postoperative RRT was 1.0% (8/817), and the incidence rate of PHLF was 10.3% (84/817). The average length of hospital stay was 20.69 days, the ICU admission rate was 7.5% (61/817), and the one-year and overall mortality were 7.2% (59/817) and 33.3% (272/817), respectively (Table 1). The median follow-up durations for determining the overall mortality and postoperative time after RRT were 3.41 (1.84 to 5.16) years and 7.00 (4.50 to 12.75) days, respectively.

### 3.1. Primary Outcomes

In the multivariate analysis, high preoperative PNI was significantly associated with a lower incidence of postoperative AKI (odds ratio (OR): 0.92, 95% confidence interval (CI): 0.85 to 0.99, *p* = 0.021). Additionally, DM (OR: 2.77, 95% CI: 1.16 to 6.58, *p* = 0.022) and the use of synthetic colloids (OR: 1.99, 95% CI: 1.05 to 3.80, *p* = 0.036) were significantly associated with a high incidence of postoperative AKI (Table 2). 

### 3.2. Secondary Outcomes

In the Cox regression analysis of one-year mortality, high preoperative PNI was significantly associated with low one-year mortality (HR: 0.87, 95% CI: 0.81 to 0.94, *p* < 0.001; Table 3). Moreover, TNM stages 3, 4A, and 4B (HR: 3.67, 95% CI: 1.14 to 11.81, *p* = 0.030 in stage 3; HR: 8.85, 95% CI: 4.29 to 18.22, *p* < 0.001 in stage 4A; HR: 9.26, 95% CI: 3.42 to 25.11, *p* < 0.001 in stage 4B) were significantly associated with high one-year mortality (Table 3).

In the Cox regression analysis of five-year mortality, high preoperative PNI was significantly associated with low five-year mortality (HR: 0.93, 95% CI: 0.90–0.97, *p* < 0.001; Table 4). Moreover, MELD scores (HR: 1.16, 95% CI: 1.03–1.30, *p* = 0.015; Table 4), TNM stages 2, 4A, and 4B (HR: 1.91, 95% CI: 1.07–3.40, *p* = 0.028 in stage 2; HR: 3.73, 95% CI: 2.72–5.11, *p* < 0.001 in stage 4A; HR: 3.35, 95% CI: 1.80–6.22, *p* < 0.001 in stage 4B), and synthetic colloid use (HR: 1.48, 95% CI: 1.08–2.03, *p* = 0.015; Table 4) were significantly associated with a high five-year mortality (Table 4).

The preoperative PNI was significantly associated with the incidence of postoperative RRT (OR: 0.76, 95% CI: 0.60 to 0.98, *p* = 0.032) and overall mortality (HR: 0.87, 95% CI: 0.79 to 0.97, *p* = 0.010), even after adjusting for other potentially confounding variables (Table 4). However, there were no significant associations between the PNI and PHLF and ICU admission (Table 5).

Figure 2 shows the Kaplan–Meier curve according to the preoperative PNI cutoff value < 45. The one-year and five-year mortality was significantly higher in the PNI < 45 group than in the PNI ≥ 45 group (log-rank test; *p* < 0.001).

## 4. Discussion

Our study demonstrated that high preoperative PNI was significantly associated with a lower incidence of postoperative AKI in patients who had undergone open hepatectomy for HCC. In addition, high preoperative PNI was significantly associated with outcomes such as low postoperative RRT requirement, low one-year, five-year and overall mortality. This suggests that the preoperative PNI might be a predictive factor for postoperative AKI and mortality in HCC patients.

Postoperative AKI in HCC patients is closely related to postoperative mortality [9], and a recent study reported postoperative AKI as the strongest independent predictor of postoperative mortality [8]. Patients with HCC are at high risk of developing postoperative AKI due to the strong association between renal and hepatic dysfunction [2]. In particular, liver cirrhosis is closely related to AKI as it induces a reduction in the effective blood volume due to dilation of the splenic and peripheral blood vessels, leading to systemic hypoperfusion and compensatory production of antidiuretic hormones [26,27]. Therefore, many efforts have been made to identify the risk factors for postoperative AKI in HCC patients [9,10,11]. Slankamenac et al. suggested that blood transfusions, hepaticojejunostomy, and oliguria were the best predictors of AKI after liver surgery [10]. Another study reported preoperative eGFR and hypertension as perioperative risk factors for AKI after liver resection surgery [11]. Lim et al. reported that an increased MELD score, major hepatectomy, and prolonged operation time were risk factors for AKI in patients who underwent hepatectomy for HCC [8]. Recently, biological markers such as the AST/ALT ratio or gamma-glutamyl transferase/alanine aminotransferase ratio (GGT/ALT) have been suggested as predictors of AKI after liver surgery [12,13].

The PNI, an index calculated using the serum albumin level and lymphocyte count, is another biological marker that indicates a patient’s nutritional and immune status [15] and is associated with various postoperative complications [28,29,30,31]. Min et al. suggested that the modified PNI predicted postoperative AKI within one week in patients receiving living donor liver transplantation, which is better than the predictive ability of the conventional MELD score [32]. To our knowledge, this is the first study to evaluate the association between the preoperative PNI and postoperative AKI in patients with HCC. In this study, a significant association was observed between the PNI and the incidence of postoperative AKI and RRT. These results suggest that the preoperative PNI may be another biological marker that predicts postoperative renal function.

In the multivariate logistic regression analysis, DM and the use of synthetic colloids were also found to be significantly associated with postoperative AKI. DM has been reported as a risk factor for AKI in surgical patients [33,34]. DM is associated with post-ischemic microangiopathy, interstitial inflammation [34], and conditions that can cause AKI even in the absence of chronic kidney disease [33].

The use of synthetic colloids such as hydroxyethyl starch in the ICU and noncardiac surgery has been associated with AKI [35,36]. The mechanisms of synthetic colloid-induced postoperative AKI are yet to be determined incomplete. However, some experimental models and clinical studies indicate that synthetic colloids can no longer be considered safe [37].

In this study, the preoperative PNI was significantly associated with postoperative AKI; this could be attributed to the unique characteristics of the PNI, a combination of albumin and lymphocyte levels, which reflects both the nutritional status and immune function of the patient. Albumin is reported to have reno-protective properties; it was associated with increased renal perfusion in an animal model [38] and closely associated with proximal tubular homeostasis [39,40] and relief of nephrotoxic effects of drugs in a human study [41]. Lymphocytes are also known to play an important role in the initiation, proliferation, and recovery of AKI [42], and one study showed a significant association between preoperative lymphocytopenia and postoperative AKI in cardiac surgery [43]. Recently, the newly identified renal T lymphocytes have been reported to have complex functions, such as a potential anti-inflammatory role in AKI [44].

Our study demonstrated that the preoperative PNI was significantly associated with outcomes such as one-year and overall mortality, which is consistent with the results of previous studies on patients who underwent hepatectomy [45,46,47]. Low serum albumin is common in chronic liver disease and is associated with a low survival rate [48]. A recent randomized clinical trial demonstrated that long-term albumin administration in patients with decompensated cirrhosis improved the overall survival and complications [49]. The lymphocyte count was significantly associated with the overall survival in HCC patients. The lymphocyte/monocyte ratio reflects the immune status of the tumor microenvironment and has been reported to be an independent predictor of survival in HCC patients [50].

There are some limitations to our study. First, our study is retrospective in nature; thus, it is possible that one of the biggest drawbacks of retrospective studies, that is, unconsidered confounding factors, triggered potential biases. However, statistical analysis was performed to minimize the impact of these confounding factors by adjusting for variables that could potentially affect the outcomes. Second, our study mostly included data from a single ethnic group within Korea; hence, the results may have been biased due to the involvement of homogeneous groups. Therefore, further research involving heterogeneous groups is needed. Third, to date, there is no exact consensus on the cutoff value of the PNI for postoperative AKI. More well-designed studies on various diseases are required for the accurate validation of preoperative PNI cutoff values that predict postoperative AKI. 

## 5. Conclusions

In conclusion, high preoperative PNI was significantly associated with a lower incidence of postoperative AKI and low mortality in patients who had undergone hepatectomy for HCC. These results suggest that the preoperative PNI provides clinically useful predictive information on postoperative AKI and surgical prognosis in HCC patients.

## Figures and Tables

**Figure 1 jpm-11-00428-f001:**
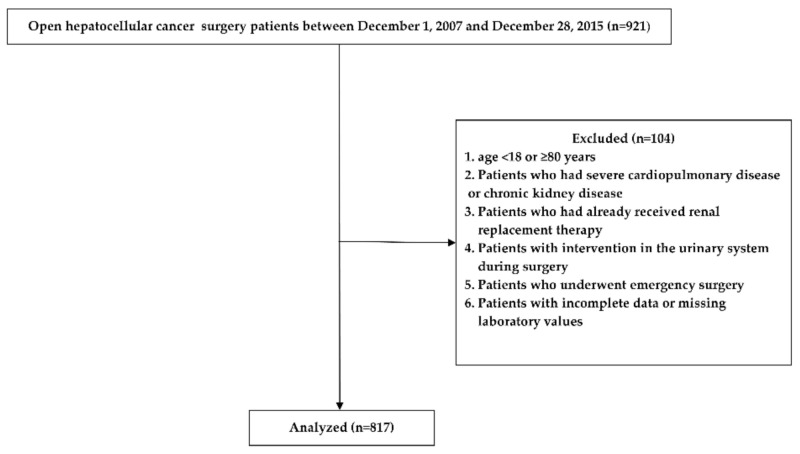
Study flow chart.

**Figure 2 jpm-11-00428-f002:**
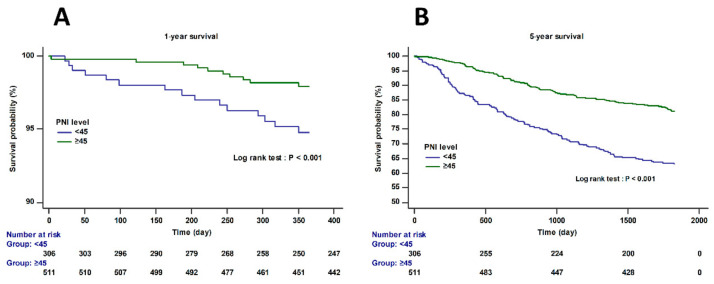
Kaplan–Meier curves for one-year (**A**) and five-year survival (**B**) according to the preoperative PNI cutoff value < 45 (log-rank test; *p* < 0.001). The proportional hazards assumption was satisfied in Schoenfeld’s residual test (*p* = 0.1828).

**Table 1 jpm-11-00428-t001:** Demographic and perioperative variables of the study population.

	Study Population (*n* = 817)
Preoperative variables	
Age; year	62.60 ± 11.83
Sex; male	512 (57.9)
Weight; kg	59.56 ± 9.84
BMI; kg m^−2^	22.90 ± 3.17
TNM staging	
1	489 (59.9)
2	48 (5.9)
3A	16 (2.0)
3B	5 (0.6)
4A	233 (28.5)
4B	26 (3.2)
Number of tumors	
Solitary	715 (87.5)
≥2	102 (12.5)
Tumor size; cm	4.99 ± 4.02
Lymph node invasion	
0	567 (69.4)
≥1	250 (30.6)
Metastasis	29 (3.5)
DM	50 (6.1)
HTN	56 (6.9)
CAD	6 (0.7)
CVA	2 (0.2)
Liver cirrhosis	298 (36.5)
ICG R15 (*n* = 731) *	
<10	233 (31.9)
10–30	476 (65.1)
≥30	20 (2.7)
MELD scores	7.19 ± 1.11
CTP scores	5.29 ± 0.49
Laboratory variables	
White blood cell	5.46 ± 1.81
Hemoglobin	13.94 ± 1.59
Platelet	166.81 ± 70.17
Prothrombin time	1.04 ± 0.08
Albumin (g·dL^−1^)	3.78 ± 0.40
Creatinine (mg·dL^−1^)	0.82 ± 0.16
eGFR (mL/min/1.73 m^2^)	78.88 ± 8.86
Total bilirubin	0.80 ± 0.38
AST	42.24 ± 34.57
ALT	38.35 ± 29.14
Sodium	139.72 ± 2.56
PNI	46.76 ± 5.64
Intraoperative variables	
Operation time (min)	269.18 ± 75.08
Type of liver resection	
Minor surgery	515 (63.0)
Major surgery	302 (37.0)
Total fluids (mL/kg)	40.09 ± 19.58
Crystalloids (mL/kg)	34.96 ± 16.76
Colloids (mL/kg)	5.13 ± 6.05
Colloid use	446 (54.6)
RBC transfusion	64 (7.8)
Units of infused RBC	0.26 ± 1.29
Urine output (mL/kg/h)	1.77 ± 1.16
Surgical outcomes	
Postoperative AKI	59 (7.2)
grade 1	52 (6.4)
grade 2	4 (0.5)
grade 3	3 (0.4)
Postoperative RRT	8 (1.0)
PHLF	84 (10.3)
Hospital stays	20.69 ± 13.07
ICU admission	61 (7.5)
One-year mortality	59 (7.2)
Five-year mortality	209 (25.6)
Overall mortality	272 (33.3)

* The total number is 731 due to missing data. BMI: body mass index; DM: diabetes mellitus; HTN: hypertension; CAD: coronary artery disease; CVA: cerebrovascular accident; ICG R15, indocyanine green retention rate at 15 min; MELD: model for end-stage liver disease; CTP: Child–Turcotte–Pugh; eGFR: estimated glomerular filtration rate; AST: aspartate aminotransferase; ALT: alanine aminotransferase; PNI: prognostic nutritional index; RBC: red blood cell; AKI: acute kidney injury; RRT: renal replacement therapy; PHLF: post-hepatectomy liver failure; ICU: intensive care unit. Values are expressed as mean ± standard deviation, median (interquartile range), or n (proportion).

**Table 2 jpm-11-00428-t002:** Univariate and multivariate logistic regression analyses of acute kidney injury.

	Univariate			Multivariate		
	OR	95% CI	*p*-Value	OR	95% CI	*p*-Value
PNI	0.93	0.89–0.98	0.004	0.92	0.85–0.99	0.021
Age (years)	1.02	0.99–1.04	0.261	1.01	0.98–1.04	0.496
Sex (male)	1.81	0.76–4.30	0.178	2.06	0.81–5.26	0.130
BMI	1.07	0.98–1.17	0.125	1.09	0.98–1.21	0.098
DM	2.67	1.19–6.00	0.017	2.77	1.16–6.58	0.022
HTN	1.60	0.66–3.91	0.300			
MELD scores	1.19	0.96–1.47	0.116	1.01	0.79–1.30	0.925
CTP scores	1.52	0.93–2.47	0.090	0.92	0.46–1.83	0.803
TNM staging			0.470			0.653
1	1.00 (Ref.)			1.00 (Ref.)		
2	1.71	0.68–4.28	0.254	1.35	0.51–3.57	0.548
3	0.65	0.15–2.79	0558	0.54	0.12–2.44	0.425
4A	0.67	0.34–1.30	0.237	0.55	0.27–1.14	0.111
4B	1.09	0.25–4.80	0.913	0.63	0.13–3.09	0.570
Operation time (min)	1.01	1.00–1.01	< 001	1.00	1.00–1.01	0.061
Type of liver resection			0.957			
Minor surgery	1.00 (Ref.)					
Major surgery	1.02	0.59–1.76				
Total fluids (mL/kg)	1.02	1.00–1.03	0.005	1.00	0.98–1.02	0.778
Synthetic colloid use	2.16	1.21–3.87	0.009	1.99	1.05–3.80	0.036
Urine output (mL/kg/h)	0.82	0.63–1.07	0.152			
RBC transfusion	1.20	1.06–1.36	0.005	1.18	0.45–3.08	0.737
Albumin (g·dL^−1^)	0.33	0.17–0.62	< 001			

OR: odds ratio; CI: confidence interval; PNI: prognostic nutritional index; BMI: body mass index; DM: diabetes mellitus; HTN: hypertension; MELD: model for end-stage liver disease; CTP: Child–Turcotte–Pugh; RBC: red blood cell. Values are expressed as mean ± standard deviation, median (interquartile range), or n (proportion).

**Table 3 jpm-11-00428-t003:** Cox regression analyses of one-year mortality.

	Univariate			Multivariate		
	HR	95% CI	p-Value	HR	95% CI	p-Value
PNI	0.85	0.81–0.89	<001	0.87	0.81–0.94	<001
Age (years)	1.00	0.97–1.02	0.916			
Sex (male)	1.00	0.97–1.02	0.916			
BMI	0.89	0.81–0.97	0.011	0.96	0.87–1.06	0.422
DM	1.19	0.43–3.28	0.738			
HTN	0.70	0.22–2.22	0.547			
MELD scores	1.19	0.98–1.45	0.088	0.94	0.74–1.20	0.635
CTP scores	2.45	1.70–3.52	<001	0.97	0.49–1.94	0.936
TNM staging			<001			<001
1	1.00 (Ref.)			1.00 (Ref.)		
2	2.35	0.52–10.65	0.270	2.36	0.52–10.79	0.270
3	3.92	1.23–12.55	0.022	3.67	1.14–11.81	0.030
4A	10.37	5.13–20.96	<001	8.85	4.29–18.22	<001
4B	19.14	7.58–48.30	<001	9.26	3.42–25.11	<001
Operation time (min)	1.00	1.00–1.00	0.028	1.00	1.00–1.00	0.713
Total fluids (mL/kg)	1.02	1.00–1.03	<001	1.00	0.99–1.02	0.548
Synthetic colloid use	0.88	0.51–1.54	0.662			
Urine output (mL/kg/h)	1.07	0.87–1.31	0.515			
RBC transfusion	1.29	1.19–1.41	<001	1.03	0.44–2.43	0.948

HR: hazard ratio; CI: confidence interval; PNI: prognostic nutritional index; BMI: body mass index; DM: diabetes mellitus; HTN: hypertension; MELD; model for end-stage liver disease; CTP: Child–Turcotte–Pugh; RBC: red blood cell. Values are expressed as mean ± standard deviation, median (interquartile range), or n (proportion).

**Table 4 jpm-11-00428-t004:** Cox regression analyses of five-year mortality.

	Univariate			Multivariate		
	HR	95% CI	*p*-Value	HR	95% CI	*p*-Value
PNI	0.91	0.88–0.93	<001	0.93	0.90–0.97	<001
Age (years)	0.99	0.98–1.00	0.196			
Sex (male)	1.00	0.69–1.44	0.995			
BMI	0.93	0.89–0.98	0.004	0.97	0.92–1.02	0.238
DM	1.20	0.71–2.02	0.505			
HTN	0.78	0.44–1.40	0.411			
MELD scores	1.22	1.10–1.35	<001	1.16	1.03–1.30	0.015
CTP scores	1.80	1.42–2.28	<001	0.96	0.68–1.35	0.824
TNM staging			<001			<001
1	1.00 (Ref.)			1.00 (Ref.)		
2	2.02	1.14–3.57	0.016	1.91	1.07–3.40	0.028
3	2.04	1.09–3.84	0.027	1.87	0.99–3.53	0.056
4A	3.47	2.57–4.70	<001	3.73	2.72–5.11	<001
4B	7.28	4.18–12.67	<001	3.35	1.80–6.22	<001
Operation time (min)	1.00	1.00–1.01	<001	1.00	1.00–1.00	0.387
Total fluids (mL/kg)	1.02	1.01–1.02	<001	1.01	1.00–1.02	0.088
Synthetic colloid use	1.32	1.00–1.74	0.048	1.48	1.08–2.03	0.015
Urine output (mL/kg/h)	1.15	1.04–1.28	0.010	1.01	0.90–1.14	0.873
RBC transfusion	3.67	2.60–5.20	<001	1.61	1.00–2.60	0.053

HR: hazard ratio; CI: confidence interval; PNI: prognostic nutritional index; BMI: body mass index; DM: diabetes mellitus; HTN: hypertension; MELD; model for end-stage liver disease; CTP: Child–Turcotte–Pugh; RBC: red blood cell. Values are expressed as mean ± standard deviation, median (interquartile range), or n (proportion).

**Table 5 jpm-11-00428-t005:** AKI incidence and surgical outcomes adjusted by PNI.

	Univariate		Multivariate *	
	OR (95% CI)	*p*-Value	OR (95% CI)	*p*-Value
AKI	0.93 (0.89–0.98)	0.004	0.92 (0.85–0.99)	0.021
Postoperative RRT	0.82 (0.73–0.93)	0.002	0.76 (0.60–0.98)	0.032
PHLF	0.88 (0.84–0.91)	<0.001	0.94 (0.88–1.00)	0.065
ICU admission	1.02 (0.97–1.06)	0.519	1.05 (0.99–1.12)	0.120
	**HR (95% CI)**	***p*** **-Value**	**HR (95% CI)**	***p*** **-Value**
One-year mortality	0.85 (0.81–0.89)	<001	0.87 (0.81–0.94)	<001
Five-year mortality	0.91 (0.88–0.93)	<001	0.93 (0.90–0.97)	<001
Overall mortality	0.91 (0.85–0.96)	0.002	0.87 (0.79–0.97)	0.010

* Adjusted for age, sex, body mass index, diabetes mellitus, hypertension, model for end-stage liver disease score, Child–Turcotte–Pugh score, TNM staging, operation time, total fluids, synthetic colloid use, and red blood cell transfusion. AKI: acute kidney injury; PNI: prognostic nutritional index; OR: odds ratio; HR: hazard ratio; CI: confidence interval; RRT; renal replacement therapy; PHLF: post-hepatectomy liver failure; ICU: intensive care unit. Values are expressed as mean ± standard deviation, median (interquartile range), or n (proportion).

## Data Availability

The dataset used and/or analyzed during the current study is available from the corresponding author on reasonable request.

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
