# Peer review of "Association of Preoperative Prognostic Nutritional Index and Postoperative Acute Kidney Injury in Patients Who Underwent Hepatectomy for Hepatocellular Carcinoma"

_jpm, 2021, doi:10.3390/jpm11050428_

Round 1
Reviewer 1 Report
In the present manuscript, Sim et al. investigated the relationship between preoperative prognostic nutritional index (PNI) and postoperative acute kidney injury (AKI). They found positive association between the two variables, and in addition, PNI was significantly related to 1-year mortality.
1. The authors stated that PNI is an index of a patient’s nutritional and immune status. However, serum albumin level is also an indicator of liver functional reserve. Therefore, the interpretation of the results needs cation. The results may only demonstrate that poor liver functional reserve was associated with postoperative morbidity and with poor survival outcomes.
2. The authors used MELD score and CTP score as indicators of liver functional reserve. However, as patients undergoing hepatic resections have relatively good liver functional reserve, these indicators have low discrimination ability among these “operable” patients. ICGR15 or other combinational indices may be useful to discriminate liver function among operable patients. How many percent of the patients were cirrhotic?
3. We routinely use diuretics after hepatectomies of hepatocellular carcinoma in cirrhotic livers. The authors are requested to describe the routine postoperative managements at their institution.
Author Response
Responses to Reviewer #1
- The authors stated that PNI is an index of a patient’s nutritional and immune status. However, serum albumin level is also an indicator of liver functional reserve. Therefore, the interpretation of the results needs cation. The results may only demonstrate that poor liver functional reserve was associated with postoperative morbidity and with poor survival outcomes.
Response: Thank you for your comments. We agree with your opinion that albumin is an indicator of liver function reserve, and PNI is a concept that includes albumin, so PNI also reflects liver function reserve. However, PNI reflects not only albumin but also lymphocytes that play an important role in adaptive immunity. These properties make it an indicator that differentiates it from albumin. In many previous studies of HCC patients, PNI has been reported to reflect not only nutritional status but also systemic inflammatory response.
- Pinato DJ, North BV, Sharma R. A novel, externally validated inflammation-based prognostic algorithm in hepatocellular carcinoma: the prognostic nutritional index (PNI). Br J Cancer. 2012;106:1439-1445.
- Wang D, Hu X, Xiao L, et al. Prognostic Nutritional Index and Systemic Immune-Inflammation Index Predict the Prognosis of Patients with HCC. J Gastrointest Surg. 2021;25:421-427.
Onodera's prognostic nutritional index has been reported as a strong prognostic indicator even for patients with hepatocellular carcinoma after initial hepatectomy, especially those with preserved liver function.
- Tanemura A, Mizuno S, Hayasaki A, et al. Onodera's prognostic nutritional index is a strong prognostic indicator for patients with hepatocellular carcinoma after initial hepatectomy, especially patients with preserved liver function. BMC Surg. 2020;20:261.
- The authors used MELD score and CTP score as indicators of liver functional reserve. However, as patients undergoing hepatic resections have relatively good liver functional reserve, these indicators have low discrimination ability among these “operable” patients. ICGR15 or other combinational indices may be useful to discriminate liver function among operable patients. How many percent of the patients were cirrhotic?
Response: Thank you for the assessment. As per your recommendation, we added ICGR15 index to Table 1. The mean ICGR15 was 13.5 %, and 233 (31.9%), 476 (65.1%), and 20 (2.7%) patients had ICGR15 values of <10%, 10–30%, and ≥30%, respectively. Also, we added the proportion of patients with liver cirrhosis (36.5%, 298/817).
- We routinely use diuretics after hepatectomies of hepatocellular carcinoma in cirrhotic livers. The authors are requested to describe the routine postoperative managements at their institution.
Response: Thanks for pointing out an important issue. Crystalloids and colloids are administered appropriately to maintain fluid balance and normal kidney function, and we try to maintain a net input/output of 0. Management with sodium restriction and judicious use of diuretic therapy is carried out when there is new onset postoperative ascites, and we do not use diuretics routinely. Blood tests are performed daily for 3 days after surgery to measure the patient's complete blood count, biochemical and electrolyte levels, and inflammation levels. We routinely use postoperative intravenous patient-controlled analgesia (PCA) for pain control. In some cases, hepatoprotective agents, such as branched-chain amino acids (BCAA) are administered to support protein synthesis and regeneration of the remnant liver. We have added sentences to the Materials and Methods section as follows: “After surgery, crystalloids and colloids were administered appropriately to maintain fluid balance and normal kidney function. Management with sodium restriction and judicious use of diuretic therapy were carried out when there was new onset postoperative ascites, and we did not use diuretics routinely. Blood tests were performed daily for 3 days after surgery to measure the patient's complete blood count, biochemical and electrolyte levels, and inflammation levels. We routinely used postoperative intravenous patient-controlled analgesia (PCA) for pain control. In some cases, hepatoprotective agents such as branched-chain amino acids (BCAA) were administered to support protein synthesis and regeneration of the remnant liver.” (page 2, lines 93-101).

Reviewer 2 Report
In this paper, Sim et al. present a very interesting large retrospective analysis of patients with HCC treated by hepatectomy, focusing on the relationship of preoperative prognostic biomarker (PNI) and postoperative acute renal failure. The authors should be congratulated for their good clinical outcomes and analysis on an important topic. Here are a couple of comments aiming to improve the manuscript further:
Minor:
- The title could be shorter… “Impact of the preoperative prognostic nutritional index on acute renal failure after hepatectomy for hepatocellular carcinoma: a retrospective study" might have been a proposal
- Please consider a scan of your manuscript with grammar and orthographic corrector for minor spelling.
Major :
- All the comments are listed in the margin of your manuscript (word file enclosed)
This study has many positive points and represents undeniable originality. However, the flaws described in the major points about methodology, and results do not allow the conclusions to be considered in a relevant way and without significant bias.
With some improving efforts considering the comments this manuscript could bring great research value.
Again, I enjoyed reading this excellent and important manuscript!

Author Response
Responses to Reviewer #2
- The title could be shorter… “Impact of the preoperative prognostic nutritional index on acute renal failure after hepatectomy for hepatocellular carcinoma: a retrospective study" might have been a proposal
Response: Thank you for your kind suggestion. However, the purpose of our study was to investigate the association between the preoperative prognostic nutritional index and postoperative acute kidney injury, so we want to keep the current title, “Association of preoperative prognostic nutritional index and postoperative acute kidney injury in patients who underwent hepatectomy for hepatocellular carcinoma.”
- Please consider a scan of your manuscript with grammar and orthographic corrector for minor spelling.
Response: We appreciate your comment. We have corrected the spelling errors in the revised manuscript.
- All the comments are listed in the margin of your manuscript (word file enclosed)
Response: All our responses to your comments are listed in the revised manuscript (word file enclosed).
- Is this number related to a power calculation related to your outcomes? If so, please add some explanation of your methodology for the power calculation.
Response: Thank you for asking. We did not calculate a sample size for this retrospective observational study; there were 817 patients who underwent open hepatectomy during the study period. This number was not related to a power calculation associated with outcomes.
- Why this particular period? Considering your 1 year mortality analysis, why didn’t you consider the last 5 years?
Response: Thank you for your comments. There was no specific reason for considering this period. As per your recommendation, we have added Cox regression analysis of 5-year mortality to Table 4 and a KM curve to Figure 2.
- For what purpouse ? Did you calculated preoperative and postoperative PNI ?
Response: At our medical center, blood tests are routinely performed as preoperative tests. Postoperative PNI was not calculated for this study.
- Could you be more specific? Do you mean the surgery ?
Response: Yes, as you may have guessed, treatment means surgery. We have clarified this in the revised manuscript as follows: “The total blood counts of all patients were determined preoperatively, < 2 days after admission, and prior to surgery.” (page 3, line 118).
Response: In our study, the primary outcome was AKI, and only outcomes, such as postoperative RRT, PHLF, ICU admission, and mortality were analyzed.
- Could you please be more extensive about this precise cut-off. Is there any rational for it ?
Response: Since Onodera suggested the cutoff PNI value of 45, this value has been frequently used. In previous studies, the PNI cutoff value for defining undernutrition has varied between 45 and 50. In our study, the cut off value for 1-year mortality through ROC analysis was 45.5 (AUC: 0.731, sensitivity: 74.58, specificity: 61.21). Therefore, we set the PNI cutoff value to 45. We have added the following sentence to the “Statistical analysis” subsection: “We set the PNI cutoff value for mortality rate to 45, guided by receiver operating characteristic analysis (AUC: 0.731, sensitivity: 74.58, specificity: 61.21).” (page 3, lines 143-144).
- Median follow-up should be reported in your tab, as well as post-operative infection and Time of RRT
Response: The median follow-up durations for determining overall mortality and postoperative time after RRT were 3.41 (1.84-5.16) years and 7.00 (4.50-12.75) days, respectively. We have clarified this in the Results section. (page 4, lines 161-163).
- Stratifying your population according your main independent variable of interest (PNI high vs low) for the type of liver resection (minor vs major surgery), time of surgery, TNM staging would have been of great value to quickly show in a table format the comparability of your 2 groups. Even though, the severity of the liver surgery shoud have been reported in your manuscript.
Response: Thank you for your valuable comments. As per your recommendation, patients were divided into two groups according to PNI (PNI <41 and PNI ≥41), and logistic regression analysis was performed and added to Supplementary Table 1. Additionally, type of liver resection (minor vs. major surgery) was added to Table 1 and Table 2.
Response: The initial crossing curve is thought to be the result of the death of one person in the PNI ≥45 group; therefore, crossing does not seem to be of great significance because the proportional hazards assumption is satisfied as a whole (p=0.1828). We have added the following sentence to the Figure 2 legend: “The proportional hazards assumption was satisfied in Schoenfeld's residual test (p = 0.1828).” (page 9, lines 224-225).
- Is it the standard procedure in your institution ? Have you considered any laparoscopic data ?
Response: Thank you for asking. During this period, we mainly performed open hepatectomy, and laparoscopic procedures were not included in the study because not many were conducted.
- This is precisely why you might have discribed the type of liver surgery.
Response: Thank you for your comments. The type of liver surgery (minor vs. major) variable has been added to Table 2; however, this variable did not correlate significantly with AKI in our study.
mPN I= 3.4 × PNI − 0.7 × serum bilirubin (mg/dL) − 12.4 × INR − 40
- One idea per paragraph should help your reader to follow and understand your discussion and argumentation… It is a little bit confusing to read about DM and with no transition read about synthetic colloid.
Response: Thank you for your kind suggestion. As per your recommendation, we have reorganized the text so that we present one idea per paragraph.

Round 2
Reviewer 1 Report
In the revised manuscript,, the authors have appropriately responded to the questions raised by the reviewers. Now I have no additional questions.
Reviewer 2 Report
Dear colleagues,
I have read your corrected manuscript with great interest and want to congratulate you on your efforts and responsiveness.